# Short-Chain Fatty Acids Ameliorate Depressive-like Behaviors of High Fructose-Fed Mice by Rescuing Hippocampal Neurogenesis Decline and Blood–Brain Barrier Damage

**DOI:** 10.3390/nu14091882

**Published:** 2022-04-29

**Authors:** Chuan-Feng Tang, Cong-Ying Wang, Jun-Han Wang, Qiao-Na Wang, Shen-Jie Li, Hai-Ou Wang, Feng Zhou, Jian-Mei Li

**Affiliations:** 1State Key Laboratory of Pharmaceutical Biotechnology, School of Life Sciences, Nanjing University, Nanjing 210023, China; tangchuanfeng@smail.nju.edu.cn (C.-F.T.); wangcongying@smail.nju.edu.cn (C.-Y.W.); mf21300027@smail.nju.edu.cn (J.-H.W.); wangqiaona@smail.nju.edu.cn (Q.-N.W.); 2School of Food Science, Nanjing Xiaozhuang University, Nanjing 211171, China; lishenjie@njxzc.edu.cn (S.-J.L.); who1978@163.com (H.-O.W.)

**Keywords:** high-fructose diet, short-chain fatty acids, stress resilience, blood-brain barrier, neurogenesis, depressive-like behaviors

## Abstract

Excessive fructose intake is associated with the increased risk of mental illness, such as depression, but the underlying mechanisms are poorly understood. Our previous study found that high fructose diet (FruD)-fed mice exhibited neuroinflammation, hippocampal neurogenesis decline and blood–brain barrier (BBB) damage, accompanied by the reduction of gut microbiome-derived short-chain fatty acids (SCFAs). Here, we found that chronic stress aggravated these pathological changes and promoted the development of depressive-like behaviors in FruD mice. In detail, the decreased number of newborn neurons, mature neurons and neural stem cells (NSCs) in the hippocampus of FruD mice was worsened by chronic stress. Furthermore, chronic stress exacerbated the damage of BBB integrity with the decreased expression of zonula occludens-1 (ZO-1), claudin-5 and occludin in brain vasculature, overactivated microglia and increased neuroinflammation in FruD mice. These results suggest that high fructose intake combined with chronic stress leads to cumulative negative effects that promote the development of depressive-like behaviors in mice. Of note, SCFAs could rescue hippocampal neurogenesis decline, improve BBB damage and suppress microglia activation and neuroinflammation, thereby ameliorate depressive-like behaviors of FruD mice exposed to chronic stress. These results could be used to develop dietary interventions to prevent depression.

## 1. Introduction

Worldwide, depression is a leading cause of disability and years of productive life lost [1,2]. Prolonged exposure to stress has been shown to precipitate episodes of major depressive disorder, particularly in vulnerable populations [3]. During the COVID-19 pandemic, the effects of quarantine measures and illness scares increased the risk of depressive disorder [4]. Moreover, unhealthy dietary patterns are also a critical factor that cannot be ignored. Western-style dietary patterns, characterized by a high consumption of sweets, high-fat dairy products and high-fat gravy, are associated with an increased risk of depression [5,6,7]. High fructose corn syrup (HFCS) is widely used in beverages and food products, and has been linked to many serious health issues, including diabetes and obesity [8]. However, there is emerging evidence that the consequences of excessive fructose intake extend to changes in emotional regulation in rodents involved in the development of depressive disorder [9,10,11,12]. Yet the conclusions are inconsistent. Moreover, the mechanism by which high fructose intake promotes the development of depression remains unclear.

Neuroinflammation is a key factor that impairs adult hippocampal neurogenesis and causes an increased rate of depression [13]. Microglia, the resident immune cells of the brain, orchestrate neuroinflammation distinctly in depression with different polarization statuses [14]. Animal studies show that western-style diets increase circulating inflammatory markers that may cross the blood–brain barrier (BBB), activate microglia and arouse primary brain neuroinflammation [13]. Meanwhile, BBB damage through the loss of tight junction proteins occludin and claudin-5 may mediate stress susceptibility and promote depression in animals [15,16]. Moreover, adult hippocampal neurogenesis confers resilience to chronic stress [17,18], but is vulnerably impacted by nutritional factors such as high-fat and high-sugar diets [19]. Studies have shown that high fructose intake initiated in adolescence activates the hypothalamic pituitary adrenal (HPA) axis, increases neuroinflammation and promotes depressive-like behaviors in male rats [10,11]. In our previous study, mice fed with a high fructose diet (FruD) also exhibited neuroinflammation, hippocampal neurogenesis decline and BBB damage [20], which may contribute to the increased risk of depression.

More interestingly, clinical studies have demonstrated that a healthy dietary pattern characterized by a high intake of dietary fibers is apparently associated with a decreased risk of depression [5,21,22]. Dietary fibers might be degraded by resident gut microbial enzymes to produce short-chain fatty acids (SCFAs), which play a major role in gastrointestinal health [23,24,25]. Of note, an increasing number of studies focus on the key roles of SCFAs affecting nervous system disorders such as mood and cognition [26,27]. Fecal SCFAs ratios are related to depressive symptoms in young adults [28]. A rodent study showed that SCFAs supplementation could alleviate enduring alterations in anhedonia and heightened stress responsiveness, as well as stress-induced increases in intestinal permeability in mice undergoing psychosocial stress [29]. Our previous study also reveals that the pathological changes in the hippocampi of FruD mice are ameliorated by SCFAs [20], but the role in the prevention of depression has yet to be extensively investigated.

In this study, mice were fed either a standard diet or a FruD to comparatively investigate their alterations of emotional behaviors, hippocampal neurogenesis and BBB permeability under chronic stress exposure. Moreover, the beneficial effects of SCFAs on the behavioral and pathological changes in FruD animals exposed to chronic stress were also studied. Pioglitazone was used as a positive control here because it exhibits antidepressant-like effects in high fat diet-fed mice [30] and chronic mild stress mice [31].

## 2. Materials and Methods

### 2.1. Animal

Four-week-old C57BL/6J mice were purchased from GemPharmatech Co., Ltd. (Nanjing, China) and kept in a specific pathogen-free facility in the Animal Research Center of Nanjing University. Animals’ husbandry and experiment procedures were approved by the Animal Ethical and Welfare Committee of Nanjing University (Approval No: IACUC-2006017). After a one-week environmental adaptation, mice were divided into two groups (60 mice per group): the control diet (CD) group were fed a standard diet and FruD group were fed a diet containing 30% fructose for eight weeks, referring to our previous report [20]. In Experiment 1, half of the mice (30 mice per group) in CD group and FruD group were exposed to chronic stress (i.e., CD-CS group and FruD-CS group) during the last 4 weeks, respectively. In brief, mice were subjected to two or three mild stressors every day, such as stroboscopic illumination, restraint, loud noise, wet cages, fasting, day and night reversal, and tilted cage (45°), etc. In Experiment 2, 30 mice from the FruD-CS group were supplied with the vehicle SCFAs in their diet (FruD-CS + SCFAs group) or pioglitazone by gavage (FruD-CS + Pi group), respectively.

### 2.2. Behavioral Tests

Following the last stress, mice from each group were divided into two parts for behavioral tests. One part (10 mice) was used to perform an open field test (OFT) and forced swimming test (FST), and the other part (10 mice) was used to perform a sucrose preference test (SPT) and tail suspension test (TST), respectively. A 2-day rest was set between any two experiments.

#### 2.2.1. SPT

Each mouse was separated into a single cage for 3 days and then acclimated to two drinking bottles with normal water or 1% sucrose solution for 2 days. The position of the bottles was interchanged every 24 h. For the preference test, mice were deprived of water for 24 h and then provided with two bottles for the ensuing 2 h (8:00–10:00 p.m.). The total weight of each bottle was measured before and after the test to get the weight of liquid consumed. The sucrose preference was calculated as the fraction of the sucrose solution compared to the total amount of consumed liquid [32,33].

#### 2.2.2. OFT

In brief, OFT was performed in an open field box, where length, width and height are 25 × 25 × 40 cm, respectively. Nine equal-area squares make up the mouse activity area. Within 10 min, time spent and distance travelled in the arena perimeter and center, as well as total distance travelled were recorded by Supermaze software. After 10 min, mice were removed from the open field box and returned to their home cages in the housing room. After each trial, 75% ethanol was used to clean the objects and boxes to eliminate odor cues.

#### 2.2.3. FST

In the swimming adaptation stage, the day before the formal FST, the mice were placed in a swimming environment for 5 min. In the formal experiment on the second day, mice were individually placed into plastic buckets (50 cm high × 20 cm internal diameter) filled with water (23–25 °C) to a depth of 15 cm. Test sessions were taped by a video camera positioned directly in front of the cylinders. These videotapes were analyzed by a well-trained observer for the duration of mouse floating, swimming and struggling with Supermaze software during the 5 min test period. 

#### 2.2.4. TST

Mice were isolated acoustically and visually. Mice were individually suspended with adhesive tape about 1 cm from the tip of the tail (about 50 cm above the floor) in a sound-isolated room. Each mouse was partitioned for a period of 6 min to avoid interference. In the end, the time of immobile, struggling and climbing were recorded using Supermaze software, by trained investigators who were unaware of the grouping situation in a blinded manner.

### 2.3. Blood Collection and Corticosterone Measurement

Three days after the end of the behavioral tests, mice from each group were fasted overnight and anesthetized with sodium pentobarbital (50 mg/kg, i.p.), and blood was obtained by cardiopuncture. Then, blood samples were centrifuged at 2500 rpm for 10 min to collect the serum for corticosterone measurement using a corticosterone ELISA kit (KGE009, R&D Systems, Minneapolis, MN, USA).

### 2.4. Immunofluorescence Staining

Three days after the end of the behavioral tests, three mice from each group were anesthetized and transcardially perfused with ice-cold 4% paraformaldehyde. Mouse brains were excised, fixed in 4% paraformaldehyde overnight, dehydrated in 20% sucrose overnight at 4 °C, and dehydrated in 30% sucrose overnight at 4 °C. Then, the brain samples were frozen in and optimum cutting temperature compound (Sakura Finetek, Torrance, CA, USA) and coronally cut with a cryostat into 30-μm-thick sections. The frozen sections were blocked with 0.1% Triton X-100 in QuickBlock™ Blocking Buffer (P0235, Beyotime, Nanjing, China) for 1 h at room temperature, and then were incubated with primary antibodies overnight at 4 °C and incubated for 1 h at 37 °C with Alexa Fluor-conjugated secondary antibodies (Invitrogen, Carlsbad, CA, USA). Hoechst (C1018, Beyotime) was used for nuclear staining. The primary antibodies included rabbit anti-Iba1 (019-19741, Wako, Saitama, Japan), mouse anti-NeuN (ab104224, Abcam, Cambridge, UK), rabbit anti-DCX (ab77450, Abcam), rabbit anti-Nestin (ab221660; Abcam) and rat anti-GFAP (13-0300, Invitrogen). Alexa Fluor-conjugated secondary antibodies included goat anti-mouse IgG (H + L) highly cross-adsorbed secondary antibody, Alexa Fluor 647 (A-21236, Invitrogen), goat anti-rabbit IgG (H + L) cross-adsorbed secondary antibody, Alexa Fluor 488 (A-11008, Invitrogen) and Alexa Fluor 555 (A-21428, Invitrogen), donkey anti-rat IgG (H + L) cross-adsorbed secondary antibody and Alexa Fluor 594 (A-21209, Invitrogen). Images were captured using a Leica TCS SP8 confocal microscope.

### 2.5. Dextran-FITC Treatment

Three days after the end of the behavioral tests, three mice from each group were injected with 200 µL 10-kDa Fluorescein isothiocyanate-dextran (20 mg/mL, FD10S, Sigma-Aldrich, St. Louis, MO, USA) through the tail vein to visualize the BBB integrity. Fifteen minutes after the injection, each of the mice was decapitated and the brain was harvested and fixed by 4% paraformaldehyde overnight at 4 °C. The brain was dehydrated in a sucrose solution and frozen in an optimal cutting temperature compound. BBB permeability was detected by immunofluorescence as shown above.

### 2.6. Isolation of Brain Vasculature

Brain samples were isolated from the executed mice after blood collection. Brain vasculature was isolated according to the methods reported by Kelly et al. [34]. Briefly, brain was homogenized with a glass homogenizer in 3 mL of cold sucrose buffer (0.32 M sucrose, 5 mM HEPES, pH 7.4). The crude brain homogenate was centrifuged at 3000 rpm for 10 min, discarding the supernatant which contain mainly neuronal components. The dense white layer of myelin on the top of the pellet was also removed and discarded. The pellet was then resuspended in 3 mL of sucrose buffer and centrifuged at 3000 rpm for 10 min to remove the rest of the myelin. An additional 800 rpm spin was used to separate the large vessels from capillaries. The remaining pellet was washed an additional four times with 1 mL of sucrose buffer and centrifuged at 2000 rpm for 10 min. The supernatant was discarded to obtain cerebral microvessels.

### 2.7. Western Blotting

Hippocampus tissues and vessel samples were lysed in an ice-cold RIPA buffer containing phenylmethylsulfonyl fluoride (PMSF). The cleared lysates were obtained by a centrifugation at 12,000 rpm at 4 °C for 15 min. Protein concentration was determined with a bicinchoninic acid (BCA) protein assay kit (23005, Thermo Fisher Scientific, Waltham, MA, USA) and bovine serum albumin was used as a standard. Total proteins were separated by sodium dodecyl sulfate-polyacrylamide gel electrophoresis and transferred onto polyvinylidene fluoride membranes (IPVH00010, Millipore, Billerica, MA, USA). After blocking in 5% milk for 1 h at room temperature, membranes were incubated with primary antibodies overnight at 4 °C and then incubated with HRP-conjugated secondary antibodies. Primary antibodies included rabbit anti-zonula occludens-1 (ZO-1, sc-10804, Santa Cruz, CA, USA), rabbit anti-Occludin (ab168986, Abcam), rabbit anti-Clandin-5 (ab15106, Abcam), rabbit anti-IL-6 (ab233706, Abcam), rabbit anti-IL-1β (ab254360, Abcam), rabbit anti-TNF-α (ab215188, Abcam) and mouse anti-β-actin (#4970, CST, Framingham, MA, USA). The protein bands were detected by Tanon-5200 Chemiluminescence Imager (Tanon Science & Technology Co., Ltd., Shanghai, China). The relative proteins levels were detected by enhanced chemiluminescence reagent and ImageJ software (Version 1.50b, National Institutes of Health, Bethesda, MD, USA), normalized to an internal reference protein and expressed as fold change relative to the control value.

### 2.8. Statistics

Data were performed blindly and presented as mean ± SEM. Two-tailed Student’s *t* test was used for comparisons between two groups. Statistical analyses were performed with GraphPad Prism Software (Graph Pad Software, San Diego, CA, USA). Value of *p* < 0.05 was considered statistically significant (* *p* < 0.05; ** *p* < 0.01; *** *p* < 0.001).

## 3. Result

### 3.1. Depressive-like Behaviors Induced by Chronic Stress Are Aggravated in FruD Mice

The experimental schedule is shown in Figure 1A. Mice were fed either a standard diet or a diet containing 30% fructose with normal drinking water for 8 weeks, and half of them were exposed to chronic stress during the last 4 weeks. Compared with the CD group, the body weight of mice significantly increased in the FruD group, whereas it significantly decreased both in the CD-CS and FruD-CS groups (Figure 1B). Moreover, the serum corticosterone levels of the mice significantly increased in the CD-CS and FruD-CS groups, and the increase was more conspicuous in the FruD-CS group (Figure 1C). At the end of chronic stress, depressive-like behaviors were assessed using the OFT, SPT, TST and FST. In detail, in the OFT, chronic stress failed to change the spontaneous activity, including total and central distance of mice in the CD-CS or FruD-CS groups, which indicates that either FruD or chronic stress does not affect the exercise capacity of mice (Figure 1D, Appendix A). In the SPT, chronic stress caused a significant decrease of sucrose preference of mice both in the CD-CS and FruD-CS groups, and the decrease was more conspicuous in the FruD-CS group (Figure 1E). In FST, chronic stress prolonged floating time and shortened swimming time of CD-CS and FruD-CS mice, and the changes were more conspicuous in the FruD-CS group (Figure 1F,G). Moreover, in the TST, FruD-CS mice displayed increased immobile time and decreased struggling time compared with mice in FruD, as well as with the CD-CS group (Figure 1H–J).

These results suggest that the increased serum corticosterone level and depressive-like behaviors in FruD mice are significantly aggravated by chronic stress.

### 3.2. Chronic Stress Worsens the Decline of Hippocampal Neurogenesis in FruD Mice

Newborn neurons are continuously generated and migrated into the granule cell layer in the hippocampus, which makes sense to avoid mental disorders. Therefore, we investigated whether the occurrence of depressive-like behaviors was related with the impact on hippocampal neurogenesis induced by FruD and/or chronic stress. Compared with that of CD mice, the number of newborn (DCX^+^) and mature neurons significantly decreased in CD-CS, FruD and FruD-CS mice (Figure 2A–C). Newborn neurons are replenished by neural stem cells (NSCs) differentiation. We also observed the changes of hippocampal NSCs in mice exposed to FruD and/or chronic stress. As expected, the number of NSCs (Nestin^+^/GFAP^+^) in CD-CS, FruD and FruD-CS mice was also significantly reduced (Figure 2D,E). Moreover, there was a significant further decrease of newborn and mature neurons, as well as NSCs, of FruD-CS mice compared with that of CD-CS or FruD mice (Figure 2A–E), indicating that hippocampal neurogenesis in FruD mice showed substantial vulnerability to chronic stress.

Together, these results demonstrated a cumulative effect of FruD and chronic stress on hippocampal neurogenesis in mice.

### 3.3. Chronic Stress Exacerbates the Damage of BBB Integrity and the Activation of Microglia in FruD Mice

Vascular and BBB-related changes underlie stress responses and resilience in mice. Fluorescent images of FITC-dextran (10 kDa) showed that vessels in FruD-CS mice are permeant to the dextran, which is visible in the parenchymal space adjacent to the vessels (Figure 3A). Brain vasculature was isolated and the levels of junction-associated proteins, including ZO-1, claudin-5 and occludin, were examined. Compared with CD mice, CD-CS mice showed a significant decrease of claudin-5 and occludin protein expression; and FruD mice displayed a significant decrease of occludin protein expression in brain vasculature (Figure 3B–D). Moreover, in FruD-CS mice, the protein level of ZO-1 was lower than that in CD-CS mice, and protein levels of claudin-5 and occludin were lower than that in CD-CS and FruD mice (Figure 3B–D).

Microglia are known as the resident immune cells in the central nervous system (CNS), and the activation of them mediates neuroinflammatory response to BBB damage [35]. As shown in Figure 3E,F, compared with CD mice, the number of microglia (Iba1^+^) in the hippocampus significantly increased both in CD-CS, FruD and FruD-CS mice, and there was a significant increase in FruD-CS mice compared with CD-CS and FruD mice. Moreover, microglia morphology of FruD-CS mice showed enlarged cell bodies and reduced branches (Figure 3E), indicating the activation of microglia. Compared with CD mice, hippocampal protein levels of IL-1β, IL-6 and TNF-α were significantly increased in CD-CS, FruD and FruD-CS mice (Figure 3G). Moreover, there was also a significant increase of these inflammatory cytokines in FruD-CS mice compared with CD-CS and FruD mice (Figure 3G).

Thus, these results indicate that cumulative negative impacts of FruD and chronic stress cause more serious damage in BBB integrity and overactivated microglia in mice. 

### 3.4. SCFAs and Pioglitazone Ameliorate the Depressive-like Behaviors of FruD-CS Mice

During the period of chronic stress exposure, FruD mice were fed with SCFAs freely or a dose of 30 mg/kg pioglitazone orally by gavage once a day (Figure 4A). Compared with CD mice, oral treatment with SCFAs, as well as pioglitazone, did not reverse the decreased body weight of FruD-CS mice (Figure 4B), whereas it suppressed the increased corticosterone level of FruD-CS mice (Figure 4C). Moreover, FruD-CS mice treated with SCFAs or pioglitazone showed no change of the spontaneous activity in the OFT (Figure 4D, Appendix A), but showed a significant increase of sucrose preference in the SPT (Figure 4E). FruD-CS mice treated with SCFAs or pioglitazone also displayed a significant decrease of floating time and prolonged swimming time (Figure 4F,G). In addition, SCFAs and pioglitazone decreased the immobile time and enhanced the struggling time in TST of FruD-CS mice (Figure 4H–J). These results showed SCFAs and pioglitazone could ameliorate depressive-like behaviors of FruD-CS mice.

### 3.5. SCFAs and Pioglitazone Rescue Hippocampal Neurogenesis Decline and BBB Damage in FruD-CS Mice

Next, we explored whether SCFAs and pioglitazone could protect hippocampal neurogenesis and BBB integrity in FruD-CS mice. As expected, SCFAs and pioglitazone significantly reversed the decreased numbers of NSCs (Figure 5A,B), newborn neurons (Figure 5C,D) and mature neurons (Figure 5C,E) in the hippocampi of FruD-CS mice.

Moreover, SCFAs and pioglitazone reduced FITC-dextran leakage (Figure 6A) and significantly increased the protein levels of ZO-1, claudin-5 and occludin in the brain vasculature of FruD-CS mice (Figure 6B–D). Further research revealed that SCFAs and pioglitazone reduced the number of microglia, and rescued microglia morphology in FruD-CS mice (Figure 6E,F). Moreover, hippocampal protein levels of IL-1β, IL-6 and TNF-α of FruD-CS mice were significantly decreased by SCFAs, as well as pioglitazone (Figure 6G).

These results demonstrated that SCFAs and pioglitazone could rescue hippocampal neurogenesis decline and BBB damage in FruD-CS mice.

## 4. Discussion

In this study, we found that chronic stress-induced depressive-like behaviors were aggravated in mice with high fructose intake. The decreased numbers of newborn neurons, mature neurons and NSCs in the hippocampi of FruD mice were worsened by chronic stress. Furthermore, chronic stress exacerbated BBB damage with decreased expression of ZO-1, claudin-5 and occludin in the brain vasculature and overactivated microglia and increased neuroinflammation in FruD mice. SCFAs supplementation improved hippocampal neurogenesis decline, BBB damage, microglia activation and neuroinflammation, which may contribute to the amelioration of depressive-like behaviors in FruD mice exposed to chronic stress.

Chronic stress is a major risk factor for depression, whereas not all individuals develop psychopathology after chronic stress exposure [36,37]. It is important to note that individual differences among people are critical as some people exhibit vulnerability whereas others are resilient to repeated exposure to stress [3]. Growing studies have been performed to elucidate the associations between dietary patterns and depression risks. Some studies report the positive associations of unhealthy dietary patterns with depression [5,6,7], whereas some other results have been inconsistent [38,39]. In this study, we confirmed the speculation that high fructose intake increased the negative impacts of chronic stress in mice, which may contribute to the increased risk of depression. Consistent with the previous report [40], susceptible mice with high fructose intake were also characterized by elevated plasma corticosterone and increased peripheral inflammation. Meanwhile, the protective effect of healthy dietary patterns on the risk of depression has been widely accepted [5,6,7,38,39]. Dietary fiber is a crucial component of a healthy diet, with benefits on the risk of depression attributing to processes in the gut microbiota [41,42]. As the main products of dietary fiber, SCFAs were found to prevent depressive-like behaviors in mice with high fructose intake and chronic stress. These results provide new evidence of the relationship between dietary patterns and depression, and support the intervention of depression through SCFAs supplementation.

The HPA axis mediates neurobehavioral and physiological responses to stress, in which the role of hippocampal neurogenesis in stress resilience has been emphasized [18]. Adult neurogenesis displays substantial vulnerability to chronic stress due to the concentrated glucocorticoid receptors in the hippocampus [43,44]. High glucocorticoids (cortisol in human and corticosterone in rodents) negatively regulate adult neurogenesis by inducing autophagic cell death of hippocampal NSCs [17,45,46]. Exposure to acute or chronic stress, as well as to chronic corticosterone administration, reduces the numbers of newborn neurons or their survival, which is implicated in the pathogenesis of anxiety and depression [19,46]. Moreover, diets high in fat and refined sugars decreased hippocampal neurogenesis and increased serum corticosterone concentrations in animals [47]. In line with our previous study [20], the numbers of newborns, mature neurons, as well as NSCs, were significantly reduced in the hippocampi of mice with high fructose intake and were aggravated by chronic stress. SCFAs supplementation showed protective effects on hippocampal neurogenesis in mice against the double damage from high fructose intake and chronic stress. These results further reveal hippocampal neurogenesis should serve as an effective therapeutic target for depression.

The brain homeostasis is supposed to be protected from any entry of pathogenic agents and immune cells because of the BBB integrity. Stress-induce BBB damage has been shown to trigger microglia activation and neuroinflammation, which contributes to increased susceptibility to depression [16,48,49]. Here, we visualized the BBB integrity by tail vein injection of FITC-dextran and found that the degree of BBB damage accompanied by decreased expression of tight junction proteins, including ZO-1, claudin-5 and occludin, in brain vessels was most apparent in mice exposed to high fructose intake and chronic stress. Moreover, we have previously demonstrated that microglial activation and neuroinflammation in the hippocampi of high fructose-fed mice were trigged by gut dysbiosis and systemic endotoxin increase [20], and were exacerbated by chronic stress, which could be at least partly due to the damage of the BBB integrity. In turn, studies reveal that overactivated microglia impairs BBB function by secreting inflammatory cytokines and phagocytose astrocytic end-feet in mice with sustained systemic inflammation [35,50]. SCFAs can reach the brain, as they were shown to cross the BBB in a cell culture model and in rats following injection of SCFAs [51,52]. This study also showed that SCFAs supplementation also ameliorated BBB damage and suppressed microglia activation and neuroinflammation in mice exposed to high fructose intake and chronic stress. These results provide more evidence for the healthy dietary intervention to prevent depression by protecting BBB integrity.

In conclusion, unhealthy high-fructose intake combined with chronic stress leads to cumulative negative effects on hippocampal neurogenesis, BBB integrity and microglia activation that promote the development of depression-like behaviors in mice. SCFAs could rescue hippocampal neurogenesis, improve BBB damage and inhibit microglia activation and neuroinflammation, thereby preventing depressive-like behaviors in mice with high fructose intake and chronic stress. The way in which SCFAs exert neuroprotective effects may not only be due to the role of the gut–brain axis, but the direct effect may also be critical and deserves further investigation. Most notably, pioglitazone, as an anti-diabetic medication, could rescue depression-like phenotypes in mice with high fructose intake, further suggesting a tight correlation between western-style dietary patterns and the increased risk of depression. The results could be used for developing interventions aimed at promoting healthy eating to prevent depression.

## Figures and Tables

**Figure 1 nutrients-14-01882-f001:**
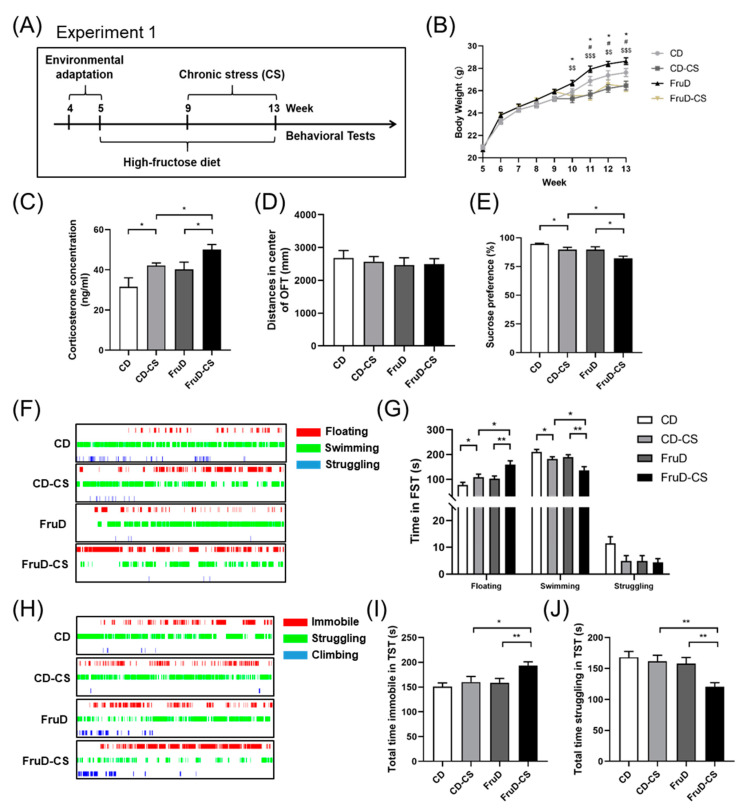
Chronic stress-induced depressive-like behaviors are aggravated in FruD mice. (**A**) Schematic diagram of mice dealing with high fructose diet and chronic stress. (**B**) Body weight of mice (* CD versus FruD group; ^#^ CD vs. CD-CS group; ^$^ FruD vs. FruD-CS group). (**C**) Corticosterone level in serum of mice. (**D**–**J**) Behavioral test and representative state images of mice in OFT (**D**), SPT (**E**), FST (**F**,**G**) and TST (**H**–**J**). Data are expressed as mean ± SEM, * *p* < 0.05, ** *p* < 0.01; ^#^
*p* < 0.05; ^$$^
*p* < 0.01, ^$$$^
*p* < 0.001.

**Figure 2 nutrients-14-01882-f002:**
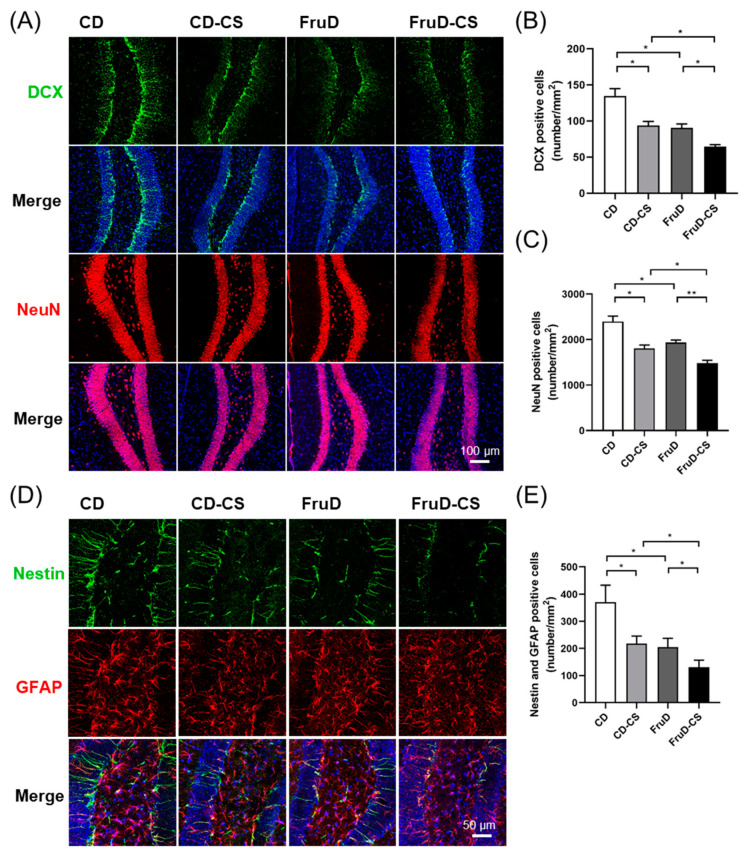
Chronic stress worsens the decline of hippocampal neurogenesis in FruD mice. (**A**) Representative confocal images labeled with DCX or NeuN. (**B**) Number of DCX positive newborn neurons was quantified. (**C**) Number of NeuN positive mature neurons was quantified. (**D**) Representative confocal images labeled with Nestin and GFAP. (**E**) Number of Nestin and GFAP positive NSCs was quantified. Data are expressed as mean ± SEM, * *p*< 0.05, ** *p*< 0.01.

**Figure 3 nutrients-14-01882-f003:**
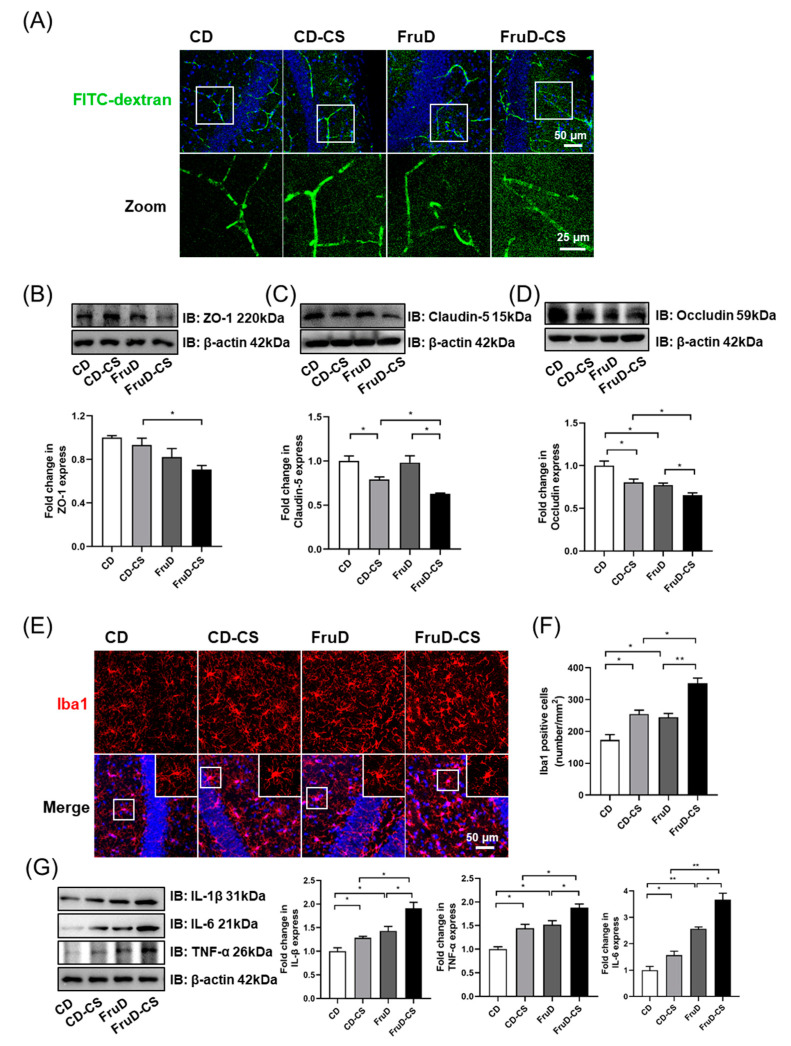
Chronic stress exacerbates the impaired BBB integrity and activates microglia in FruD mice. (**A**) Representative confocal images for brain vessel visualization in mice injected with FITC-dextran. (**B**–**D**) Representative immunoblot and quantification of ZO-1 (**B**), claudin-5 (**C**) and occludin (**D**) protein levels in the brain vessel. (**E**) Representative confocal images labeled with Iba1. (**F**) Number of Iba1 positive microglia was quantified. (**G**) Representative immunoblot and quantification of IL-1β, IL-6 and TNF-α protein levels in the hippocampus. Data are expressed as mean ± SEM, * *p* < 0.05, ** *p* < 0.01.

**Figure 4 nutrients-14-01882-f004:**
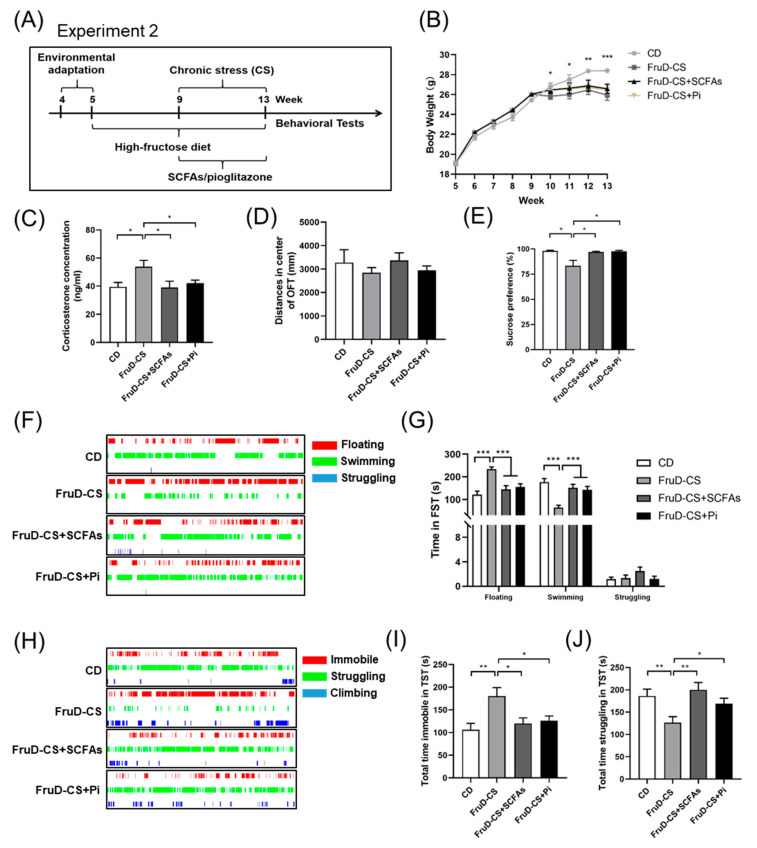
SCFAs and pioglitazone ameliorate depressive-like behaviors of FruD-CS mice. (**A**) Schematic diagram of mice dealing with SCFAs and pioglitazone treatment in FruD-CS mice. (**B**) Body weight of mice (* CD versus FruD-CS group). (**C**) Corticosterone level in serum of mice. (**D**–**J**) Behavioral test and representative state images of mice in OFT (**D**), SPT (**E**), FST (**F**,**G**) and TST (**H**–**J**). Data are expressed as mean ± SEM, * *p* < 0.05, ** *p* < 0.01, *** *p* < 0.01.

**Figure 5 nutrients-14-01882-f005:**
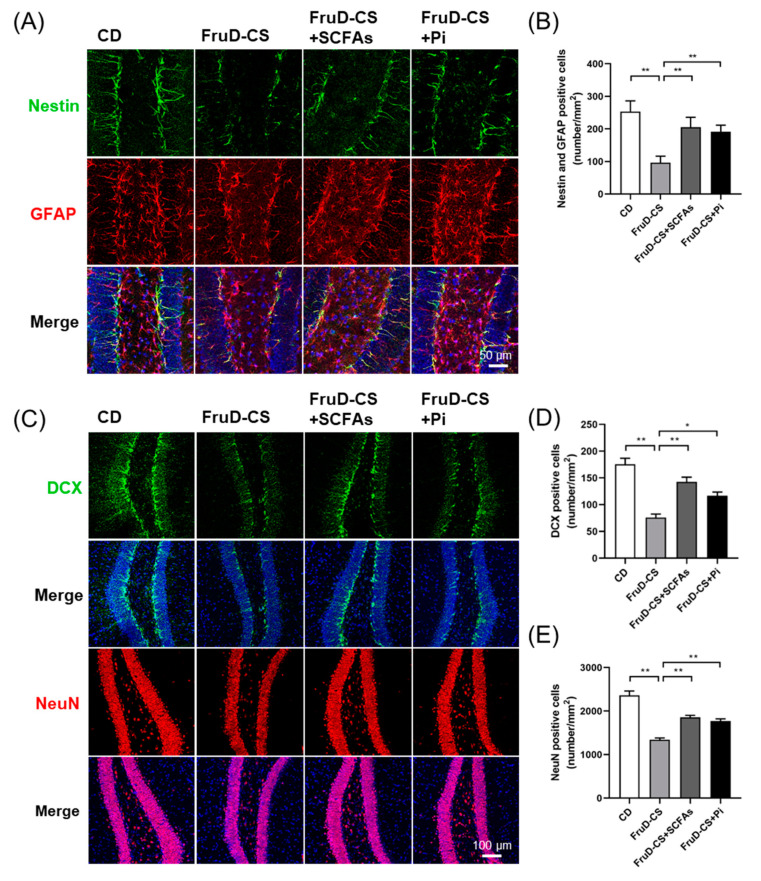
SCFAs and pioglitazone rescued the decline of hippocampal neurogenesis in FruD-CS mice. (**A**) Representative confocal images labeled with Nestin and GFAP. (**B**) Number of Nestin and GFAP positive NSCs was quantified. (**C**) Representative confocal images labeled with DCX or NeuN. (**D**) Number of DCX positive newborn neurons was quantified. (**E**) Number of NeuN positive mature neurons was quantified. Data are expressed as mean ± SEM, * *p* < 0.05, ** *p* < 0.01.

**Figure 6 nutrients-14-01882-f006:**
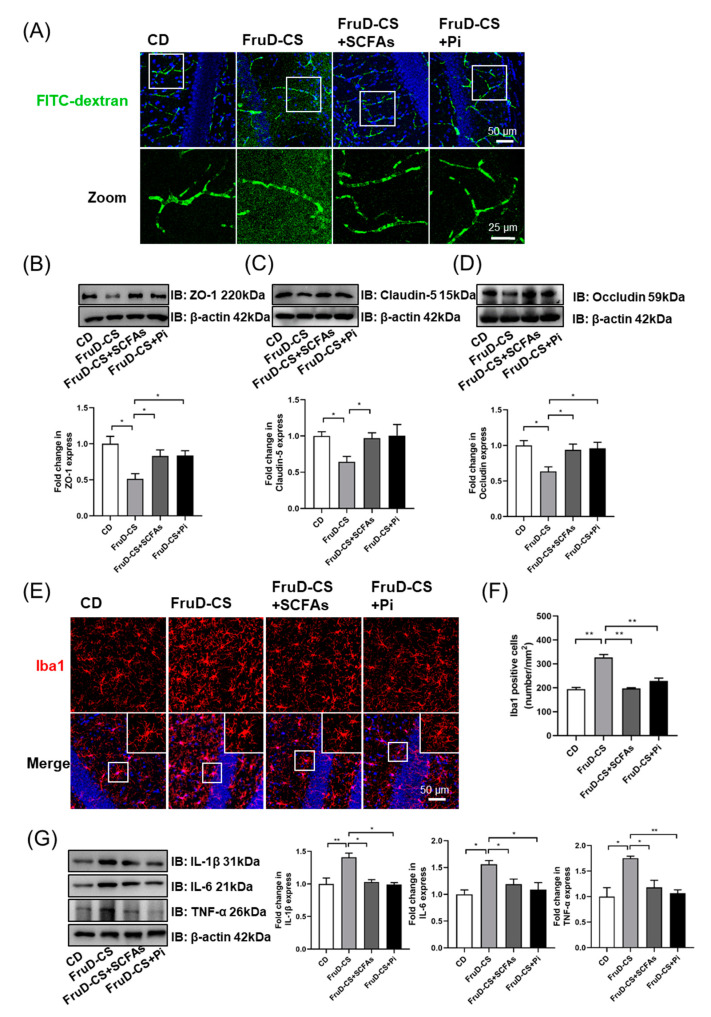
SCFAs and pioglitazone repaired BBB damage in FruD-CS mice. (**A**) Representative confocal images for brain vessel visualization in mice injected of FITC-dextran. (**B**–**D**) Representative immunoblot and quantification of ZO-1 (**B**), claudin-5 (**C**) and occludin (**D**) protein levels in brain vessel. (**E**) Representative confocal images labeled with Iba1. (**F**) Number of Iba1 positive microglia was quantified. (**G**) Representative immunoblot and quantification of IL-1β, IL-6 and TNF-α protein levels in the hippocampus. Data are expressed as mean ± SEM, * *p* < 0.05, ** *p* < 0.01.

## Data Availability

The data presented in this study are available on request from the corresponding author.

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
