# Peer review of "Short-Chain Fatty Acids Ameliorate Depressive-like Behaviors of High Fructose-Fed Mice by Rescuing Hippocampal Neurogenesis Decline and Blood–Brain Barrier Damage"

_nutrients, 2022, doi:10.3390/nu14091882_

Round 1

Reviewer 1 Report

In this manuscript, the authors claim an induced susceptibility to chronic stress in mice with a high fructose diet. They show that upon exposure to stress, mice on a HFD have a decreased hippocampal neurogenesis, increased BBB damage and over-activated microglia compared to mice with a normal diet under similar stress conditions.  

I disagree with the main conclusion of this manuscript that FruD mice are more susceptible to chronic stress. To me, what the results are suggesting is not a more pronounced effect of chronic stress in HFD mice but rather a cumulative effect of stress and HFD. Both factors on their own demonstrate similar effects so it is expected to observe a more pronounced effect when both are present at the same time. Since the authors have already investigated the effect of HFD in previous publications the main conclusion here should be related to the specific effect of chronic stress on neuroinflammation and behavior which is similar to that observed with HFD. If the authors want to claim that HFD is at the origin of a greater susceptibility to the effects of CS they should look at some mechanistic distinction between the two factors.  

Line 210-212 The authors state that no depressive-like behaviors were observed in FruD mice. This needs to be clarified. From the data, the HFD on its own has the same effect on corticosterone levels and behaviors than chronic stress. This means that both CS and HFD contribute to the greater effect observed in the HFD-CS group.

I found several grammatical mistakes throughout the manuscript which should be carefully reviewed. Some examples can be found below:

Line 38 Especially during 36 COVID-19 pandemic, the influence of COVID-19, isolation measures and disease panic et al. increases the risk of depressive disorder

Line 51 clinical studies

Line 38 Whereas doesn’t make grammatical sense

Line 74-75 , although they appear neuroinflammation, BBB damage and impaired hippocampal neurogenesis, accompanying by gut microbiome disturbance and SCFAs reduction.

Line 86 1 weeks

Line 98-99 Acclimated vs acclimatized

Line 154 as described above

Line 179 by centrifugation

What is causing the increased activation of microglia upon BBB disruption? Is it the penetration of inflammatory molecules? This should be investigated or at least discussed based on their previous findings in the gut.

The authors should also discuss how they think microglia reduce tight junction proteins expression.

Reviewer 2 Report

In this present manuscript, Chuan-Feng Tang and colleagues investigated the effects that a high-fructose diet in combination with chronic stress had on behaviour, hippocampal neurogenesis and blood brain barrier integrity. The authors have shown that mice fed with a high fructose diet displayed more depressive-like behaviors, impaired neurogenesis and damaged BBB compared with standard chow fed animals, effects that were even more pronounced when combined with chronic stress. Moreover, it is shown that short-chain fatty acids (SCFAs) supplementation on the diet prevented all these diet-induced adverse effects. Altogether the results reported in the manuscript indicate that a high-fructose diet under chronic stress might increase the probability to develop mood disorders, which can be improved by supplemental administration of SCFAs.

I would suggest the revision of the text by an English native speaker as some sentences are not very clear. Collectively, the experimental design is adequate and properly performed. However, there are some methodological aspects that the authors can address to improve the manuscript for its further publication.

  • What is the purpose of the pioglitazone administration in experiment 2? If I am not mistaken it is an anti-diabetic medication used to treat type 2 diabetes. There is no background about this kind of medications in relation to depressive-like disorders, neurogenesis etc in the introduction, so as it is being used in the experiments it would be useful to have any reference.

  • Regarding the behavioural tests section in methods, it would help to have the entire name of the test instead of the short name (i.e Sucrose preference test (SPT).

  • How were the battery of behavioral test done? Were all performed in the same order in all the mice or randomised? During how many days and how many days in between tests?

  • When was the blood collected for the corticosterone assay?

  • How were the animals chronically stressed? There is not mention to that in the methods section.

  • Regarding the immunofluorescence staining (line 136). …the slices or cells were incubated.. I believe cells could be removed.

  • Was the destran-FITC treatment done in all the mice? When was it performed? Before or after the behavioural tests? Since these brains are used for both immunofluorescence and western blot it is not described in the methods how the brains were processed for both assays.

  • In general, the figures seem to me very small, especially the experimental diagrams are very difficult to read. I would increase the size of the figure and font. Also, the immunofluorescence images are small and too dark, and it is difficult to observe any difference between the groups.

  • In section 3.3 line 251-252 states “Moreover, protein levels of zonula occludens- 1 (ZO-1), claudin-5 and occludin were lower in FruD-CS mice than that in CD-CS or FruD mice (Figure 3B-D)” However, in the graphs the symbol showing the difference between Fructose-CS and Fructose only seems to be missing.

Author Response

Re: nutrients-1679541

Dear Editors and Reviewers,

We are very glad to receive your letter that informs us to revise our manuscript (nutrients-1679541). We greatly appreciate the invaluable comments and suggestions from you and the reviewers that improve the manuscript. We have made the responses to all the comments and suggestions point by point as follows.

We thank you again for giving us the chance to revise the manuscript. We are looking forward to your reply.

Response to Reviewer #2’ comments and suggests

From Reviewer #2:

In this present manuscript, Chuan-Feng Tang and colleagues investigated the effects that a high-fructose diet in combination with chronic stress had on behaviour, hippocampal neurogenesis and blood brain barrier integrity. The authors have shown that mice fed with a high fructose diet displayed more depressive-like behaviors, impaired neurogenesis and damaged BBB compared with standard chow fed animals, effects that were even more pronounced when combined with chronic stress. Moreover, it is shown that short-chain fatty acids (SCFAs) supplementation on the diet prevented all these diet-induced adverse effects. Altogether the results reported in the manuscript indicate that a high-fructose diet under chronic stress might increase the probability to develop mood disorders, which can be improved by supplemental administration of SCFAs.

 I would suggest the revision of the text by an English native speaker as some sentences are not very clear. Collectively, the experimental design is adequate and properly performed. However, there are some methodological aspects that the authors can address to improve the manuscript for its further publication.

Re: Thank you for your question.

We have carefully checked and revised grammatical errors throughout of the manuscript. Many sentences have been rewritten to clear the points and more detail information has been provided in Method section.

 What is the purpose of the pioglitazone administration in experiment 2? If I am not mistaken it is an anti-diabetic medication used to treat type 2 diabetes. There is no background about this kind of medications in relation to depressive-like disorders, neurogenesis etc in the introduction, so as it is being used in the experiments it would be useful to have any reference.

Re: Thank you for your question.

Pioglitazone was used as a positive control here, because it exhibits antidepressant-like effects in high fat diet-fed mice [1] and chronic mild stress mice [2]. Most notably, pioglitazone, as an anti-diabetic medication, could rescue depression-like phenotypes in mice with high fructose intake and chronic stress, further suggesting a tight correlation between western-style dietary pattern and the increased risk of depression.

 Regarding the behavioural tests section in methods, it would help to have the entire name of the test instead of the short name (i.e Sucrose preference test (SPT).How were the battery of behavioral test done? Were all performed in the same order in all the mice or randomised? During how many days and how many days in between tests?

Re: Thank you for your question. We have provided the entire names, as well as the short names of behavioral tests in 2.2 section.

After the last stress, mice of each group were divided into two parts for behavioral tests. One part (15 mice) was used to perform open field test (OFT) and forced swimming test (FST), and the other part (15 mice) was used to perform sucrose preference test (SPT) and tail suspension test (TST), respectively. A 2-day rest was set between any 2 experiments. All the behavioral tests were conducted during 1week.

  When was the blood collected for the corticosterone assay?

Re: Thank you for your question.

Three days after the end of the behavioral tests, the mice were fasted over-night and anesthetized with sodium pentobarbital (50 mg/kg, i.p.), and blood was obtained by cardiopuncture. Then, blood samples were centrifuged at 2500 rpm for 10 min to collect the serum for corticosterone measurement using a corticosterone ELISA kit (KGE009, R&D Systems, USA).

 How were the animals chronically stressed? There is not mention to that in the methods section.

Re: Thank you for your question. We have provided the detail information in the method.

In brief, mice were subjected to 2 or 3 mild stressors every day, such as stroboscopic illumination, restraint, loud noise, wet cage, fasting, day and night reversal, titled cage (45°), etc.

 Regarding the immunofluorescence staining (line 136). …the slices or cells were incubated.. I believe cells could be removed.

Re: Thank you for your suggestion. We have removed “cells”.

Was the destran-FITC treatment done in all the mice? When was it performed? Before or after the behavioural tests? Since these brains are used for both immunofluorescence and western blot it is not described in the methods how the brains were processed for both assays.

Re: Thank you for your question. We have addressed these questions in the Method section. The brain samples for different test were isolated from the mice subjected different process in each group.

Three days after the end of the behavioral tests, 3 mice from each group for immunofluorescence were anesthetized and transcardially perfused with ice-cold 4% paraformaldehyde, 3 mice from each group were injected with 200 µl 10-kDa Fluorescein isothiocyanate-dextran (20 mg/ml, FD10S, Sigma, Germany) through the tail vein to visualize the BBB integrity and other mice were executed after blood collection for brain vasculature isolation and western blot assay.

 In general, the figures seem to me very small, especially the experimental diagrams are very difficult to read. I would increase the size of the figure and font. Also, the immunofluorescence images are small and too dark, and it is difficult to observe any difference between the groups.

Re: Thank you for your suggestion. We have adjusted the size and clarity of each figures in the revised manuscript.

 In section 3.3 line 251-252 states “Moreover, protein levels of zonula occludens- 1 (ZO-1), claudin-5 and occludin were lower in FruD-CS mice than that in CD-CS or FruD mice (Figure 3B-D)” However, in the graphs the symbol showing the difference between Fructose-CS and Fructose only seems to be missing.

Re: Thank you for your question. We have revised the sentence as follows.

Moreover, in FruD-CS mice, protein level of ZO-1was lower than that in CD-CS mice, and protein levels of claudin-5 and occludin were lower than that in CD-CS and FruD mice (Figure 3B-D).

  1. Lam, Y.Y.; Tsai, S.F.; Chen, P.C.; Kuo, Y.M.; Chen, Y.W. Pioglitazone rescues high-fat diet-induced depression-like phenotypes and hippocampal astrocytic deficits in mice. Biomed Pharmacother 2021, 140, 111734.
  2. Zhao, Q.; Wu, X.; Yan, S.; Xie, X.; Fan, Y.; Zhang, J.; Peng, C.; You, Z. The antidepressant-like effects of pioglitazone in a chronic mild stress mouse model are associated with PPARgamma-mediated alteration of microglial activation phenotypes. J Neuroinflammation 2016, 13, 259.

Round 2

Reviewer 1 Report

I thank the authors for addressing most of my concerns.

I would recommend carefully going through the manuscript once again to correct several remaining grammatical mistakes. Some examples include:

Line 28: ‘’…overactivated microglia and increased neuroinflammation in FruD mice.’’

Line 100: ‘’…prevention of depression has yet to be extensively investigated.’’

Line 281: ‘’…, while it was significantly decreased both….’’

Line 360-361: ‘’As shown in Figure 3E and F, ….’’

Line 393: ‘’….while it suppressed the increased….’’

Line 417: ‘’reversed the decreased numbers….’’

Line 475: ‘’..high fructose intake increased the negative impacts…’’

The conclusions to Figure 2 (Line 330-334) should be modified to reflect the main conclusion of the manuscript now being a cumulative effect of CS and HFD rather than an increased susceptibility to CS. Similarly, the conclusion at line 486 should be reformulated to avoid talking about susceptibility.

Line 524: The sentence starting by Moreover, we demonstrated… should be modified to clearly state that this is a previous finding; Moreover, we have previously demonstrated…..

The new results from Figure 3G and 6G should be incorporated in the discussion section.
